# $\mathcal{M}^4$: A Unified XAI Benchmark for Faithfulness Evaluation of Feature Attribution Methods across Metrics, Modalities and Models

**Xuhong Li**
Baidu Inc.
lixuhong@baidu.com

**Mengnan Du**
New Jersey Institute of Technology
mengnan.du@njit.edu

**Jiamin Chen**
Baidu Inc.
chenjiamin01@baidu.com

**Yekun Chai**
Baidu Inc.
chaiyekun@baidu.com

**Himabindu Lakkaraju**
Harvard University
hlakkaraju@seas.harvard.edu

**Haoyi Xiong**[*]
Baidu Inc.
xionghaoyi@baidu.com

## Abstract

While Explainable Artificial Intelligence (XAI) techniques have been widely studied to explain predictions made by deep neural networks, the way to evaluate the faithfulness of explanation results remains challenging, due to the heterogeneity of explanations for various models and the lack of ground-truth explanations. This paper introduces an XAI benchmark named $\mathcal{M}^4$, which allows evaluating various input feature attribution methods using the same set of faithfulness metrics across multiple data modalities (images and texts) and network structures (ResNets, MobileNets, Transformers). A taxonomy for the metrics has been proposed as well. We first categorize commonly used XAI evaluation metrics into three groups based on the ground truth they require. We then implement classic and state-of-the-art feature attribution methods using InterpretDL and conduct extensive experiments to compare methods and gain insights. Extensive experiments have been conducted to provide holistic evaluations as benchmark baselines. Several interesting observations are made for designing attribution algorithms. The implementation of state-of-the-art explanation methods and evaluation metrics of $\mathcal{M}^4$ is publicly available at `https://github.com/PaddlePaddle/InterpretDL`.

## 1 Introduction

Although deep neural networks (DNNs) have achieved state-of-the-art performance on numerous AI tasks, they are often regarded as black boxes due to their lack of transparency. This opacity hinders the adoption of deep models in high-stake applications that require explainability, such as healthcare, criminal justice, and law. Explainable AI (XAI) aims to address this limitation by developing techniques to provide explanations for predictions made by DNN models [15, 17]. In recent years, researchers have proposed various XAI algorithms to enable deeper understanding of DNNs [40, 46, 48, 6, 7, 9]. Among these techniques, post-hoc feature attribution is one of the most widely used paradigms, which could provide insight into the behaviors of trained DNNs.

Although these feature attribution methods can be helpful in understanding deep models, their **faithfulness** (i.e., how well explanations match model reasoning) is not always guaranteed. Unfaithful explanations fail to provide a complete or accurate description of the algorithm that the model implements, thus might yield futile or deceptive insights [41]. To address this, researchers have

---

[*]Corresponding author

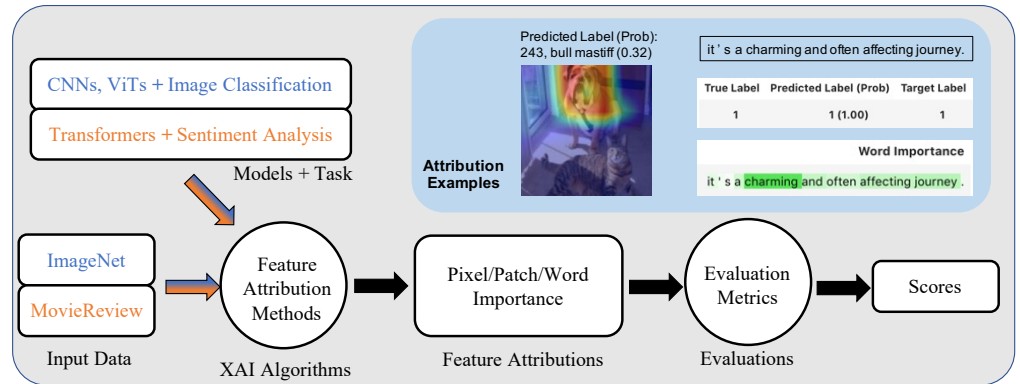

Figure 1: The benchmark pipeline $\mathcal{M}^4$, which supports evaluations on two modalities with more than ten deep models to holistically validate the faithfulness of existing feature attribution methods.

proposed methods to measure the faithfulness of explanations and filter out unfaithful algorithms. For example, Adebayo et al. [2] have proposed randomization tests on the model's parameters, on which explanation methods should depend. Such tests can easily filter out unfaithful explanation methods that rarely vary even when randomizing the model parameters. Furthermore, beyond simply binarizing the faithfulness of the explanation methods, recent work has built XAI benchmarks with evaluation metrics and datasets [14, 39, 3] to quantitatively compare the faithfulness among explanation methods. These evaluations are emerging as a guiding principle for selecting the most effective and appropriate explanation methods to elucidate models in specific tasks.

However, there are still several challenges towards this direction of research for faithfulness evaluations and benchmarks of XAI algorithms. First, assessing the faithfulness of feature attributions is difficult due to the lack of ground truths (of explanations). Although some attainable metrics have been proposed to evaluate feature attributions [42, 53], existing benchmarks rarely share or standardize these metrics. For instance, perturbation-based metrics are commonly used but differ in their perturbation methods, scales, and granularity across benchmarks. Alternative metrics also quantify faithfulness in different ways, but few studies have consolidated, classified, or analyzed the relationships between these metrics. Second, most benchmarks focus on a single data type and model, limiting their applicability and ability to validate new methods. Feature attribution techniques designed for specific models may not be applicable to others. For example, gradient-based explanations can be noisy for Vision Transformers [4]. To robustly evaluate explanation methods, XAI benchmarks should encompass assessments across multiple models and modalities.

To address these challenges, we propose developing benchmarks that cover various types of data and models, employ a taxonomy of standardized metrics, and analyze the relationships between different metrics. First, we categorize the commonly used faithfulness metrics into three groups, according to the status of the ground-truth explanations. Then, based on these metrics, we provide a unified XAI benchmark named $\mathcal{M}^4$, which supports the evaluations of two modalities (images and texts) with more than ten deep models to holistically validate the faithfulness of existing feature attribution methods (Figure 1). Through the proposed benchmark $\mathcal{M}^4$, we conducted comprehensive experiments and also obtained several interesting observations on the feature attribution methods that motivate future work in designing XAI methods and other applications related to XAI.

The contributions of this work are threefold:

- To the best of our knowledge, our work provides the first taxonomy of faithfulness evaluation metrics. We categorize the commonly used metrics into three sets that require no ground-truth of explanations, generate pseudo ground truth, and design synthetic ground truth, respectively.

- The proposed $\mathcal{M}^4$ benchmark allows the evaluation of diverse feature attribution methods with the same metrics across multiple data modalities (images and texts) and various architectures. We take advantage of the modular-designed implementations of XAI methods in InterpretDL [29] to build the $\mathcal{M}^4$ benchmark, making the evaluations easily extensible.

- We conduct extensive experiments to provide holistic comparisons with off-the-shelf baselines, yielding valuable observations that can inform future designs and applications of XAI methods.

# 2 Benchmark $\mathcal{M}^4$

In this section, we introduce $\mathcal{M}^4$, a unified benchmark for evaluating feature attribution methods.

## 2.1 Tasks, Datasets and Models

We consider two classic tasks: image classification from the computer vision domain and sentiment analysis from the NLP domain. These tasks have been used as testbeds for most explanation methods. To maximize the reuse of publicly available resources, we utilize two commonly used datasets for this benchmark, namely ImageNet [12] and MovieReview [55]. In fact, we do not need to train models on ImageNet because there are numerous pre-trained models publicly available. Furthermore, to enhance computational efficiency, we use a subset of 5,000 images from the ImageNet validation set, with 5 images per class for class balance. One training phase for images is required to quantify the Synthetic-based score (introduced in Section 2.3) because models need to be trained on a new synthetic dataset. For this training scenario, we take 10 random images per class from the ImageNet training set and randomly add synthetic patches to train the model. We then conduct the evaluations on the same 5,000 images (also with random synthetic patches) as the previous evaluations. For MovieReview, we fine-tune the pretrained language models on its training set and conduct the faithfulness evaluation on its validation set. Therefore, both the training and validation sets of MovieReview are required.

The reasons for choosing the ImageNet and MovieReview datasets, as well as the related tasks also include the availability of semantic segmentation labels [19, 31] and language reasoning labels [14] from public resources. Although these labels are not used in $\mathcal{M}^4$ as our purpose is for faithfulness evaluations, these labels can be used directly to measure human-labeled *interpretability* which is defined as the alignment between model explanations with human understanding.

Recent benchmarks rarely consider the choices of network structures when evaluating the faithfulness of explanation methods. In contrast, our proposed $\mathcal{M}^4$ considers a wide range of models to holistically evaluate faithfulness: VGG [45], three ResNets [21], Mobilenet-V3 [22], three ViT versions (small, base and large) [16] and MAE-ViT-base [20] for image classification, and two BERTs (base and large) [13], DistilBERT [43], ERNIE-2.0-base [47] and RoBERTa [34] for sentiment analysis.

## 2.2 Feature Attribution Methods

Our proposed benchmark $\mathcal{M}^4$ considers classic feature attribution methods that have been used in previous benchmarks, including model-agnostic explanations (LIME [40]), gradient-based (Integrated Gradient (IG) [48], SmoothGrad (SG) [46]), and model-specific (GradCAM [44]). In addition, our proposed benchmark $\mathcal{M}^4$ also evaluates the state-of-the-art ones especially for Transformer structures, e.g, Generic Attribution (GA) [6], Bidirectional Explanations (BT) [9], *etc.*, to comprehensively evaluate the feature attributions, as well as to revisit and confirm the progress of explanation methods.

Feature attribution methods are explainers that assign importance scores to input features of a machine learning model, elucidating the reasoning behind a model's specific prediction. Such explanations are readily comprehensible for human understanding. We note that there are more advanced forms of explanation, including prototype exemplars [8, 18, 37], concept vectors in the activation space [24, 54], and proxy models to simulate the rational process of deep models [27, 56], *etc*. Although these methods produce unique and probably more profound explanations, they may be either incomprehensible or laboriously evaluated. As an initial step toward establishing a benchmark for evaluating XAI methods, concentrating on feature attribution methods would be an achievable endeavor, albeit one that still presents its own set of challenges.

## 2.3 Metrics and Taxonomy

Due to the lack of explanation ground truths, there are no natural metrics to quantify the faithfulness of the explanations produced by feature attribution methods. Recent studies have proposed various metrics to evaluate explanation methods [42, 28, 53]. We review commonly used metrics and categorize them into three types based on whether they require explanation ground truths and how the ground truths are generated. Specifically, the three types of metrics are as follows.

**No Ground Truth**. Perturbation-based metrics offer a feasible approach that circumvents the need for ground-truth explanations. The underlying concept is based on the premise that important input

features will noticeably degrade the predicted probability of a model, while irrelevant features will have minimal impact on the probability mass. Several widely utilized metrics are built upon this fundamental idea. For example, Samek et al. [42] proposed *ordered perturbation*, which perturbs the input features gradually following the same order of values from the explanation results. They then calculate the area under the perturbation curve. The MoRF (most relevant first) metric represents the descending order, the LeRF (least relevant first) denotes the ascending order, and the ABPC (area between the perturbation curves) signifies their difference. Similarly deletion and insertion metrics [38] are an equivalent pair of MoRF and LeRF. Another example is *random perturbation*, such as Infidelity [53], which randomly perturbs the input, unlike ordered perturbation, and computes the empirical average of Eq.(4). Formally, their formulations are given as follows:

$$\text{MoRF}(\boldsymbol{x}) = \frac{1}{L+1} \sum_{k=0}^{L} (\boldsymbol{f}(\boldsymbol{x}_{\text{MoRF}}^{(0)}) - \boldsymbol{f}(\boldsymbol{x}_{\text{MoRF}}^{(k)})) \,, \tag{1}$$

$$\text{LeRF}(\boldsymbol{x}) = \frac{1}{L+1} \sum_{k=0}^{L} (\boldsymbol{f}(\boldsymbol{x}_{\text{LeRF}}^{(0)}) - \boldsymbol{f}(\boldsymbol{x}_{\text{LeRF}}^{(k)})) \,, \tag{2}$$

$$\text{ABPC}(\boldsymbol{x}) = \frac{1}{L+1} \sum_{k=0}^{L} (\boldsymbol{f}(\boldsymbol{x}_{\text{LeRF}}^{(k)}) - \boldsymbol{f}(\boldsymbol{x}_{\text{MoRF}}^{(k)})) \,, \tag{3}$$

$$\text{INFD}(\boldsymbol{x}) = \mathbb{E}_{\boldsymbol{I} \sim \mu_{\boldsymbol{I}}} (\boldsymbol{I}^T \mathcal{A}(\boldsymbol{x}, \boldsymbol{f}) - (\boldsymbol{f}(\boldsymbol{x}) - \boldsymbol{f}(\boldsymbol{x} - \boldsymbol{I}))^2), \tag{4}$$

where $\boldsymbol{f}$ is the DNN model including the architecture and the trained parameters, $\boldsymbol{x}^{(0)}$ is the original input, $\boldsymbol{x}_{\text{MoRF}}^{(k)}$ is the perturbed input whose top-$k$ features are masked, $\boldsymbol{x}_{\text{LeRF}}^{(k)}$ is the perturbed input whose bottom-$k$ features are masked, and $\mathcal{A}$ is a feature attribution method taking a data sample $\boldsymbol{x}$ and a trained model $\boldsymbol{f}$ as input. Note that if the explanation is of better quality and of better loyalty to the model, the MoRF and ABPC scores are higher and the LeRF is lower. However, the metric of ABPC scores contains the information of both MoRF and LeRF. Without loss of completeness, we do not report the results of LeRF scores. INFD [53] follows the similar idea but its perturbation manner is quite different. Random perturbation on the input space is adopted (or an effective sampling strategy can be designed). This may lead to high computational complexity when the input space is large.

**Pseudo Ground Truth**. In certain cases, pseudo ground truths can serve as reasonable approximations of the actual ground truths for explanations. For example, pseudo ground truths of explanations can be generated through a consensus-based metric [28]. Here, the consensus refers to the aggregation of explanations from multiple deep models. We can consider this consensus as a pseudo ground truth for the explanation. To evaluate an explanation, we only need to measure its similarity score to this pseudo ground truth. We call this score the PScore. Formally, PScore can be formulated as follows:

$$\text{PScore}(\boldsymbol{x}) = \cos(\frac{1}{|\mathcal{M}|} \sum_{\boldsymbol{g} \in \mathcal{M}} \mathcal{A}(\boldsymbol{x}, \boldsymbol{g}), \mathcal{A}(\boldsymbol{x}, \boldsymbol{f})) \,, \tag{5}$$

where the cosine similarity is taken, $\mathcal{M}$ is a set of well-trained models and $\mathcal{A}$ is a feature attribution method that takes a data sample $\boldsymbol{x}$ and a trained model $\boldsymbol{f}$ as input.

Some clarifications may be required for elaborating the pseudo ground truth metric.

(1) Take an example of evaluating the faithfulness of a new attribution algorithm $\mathcal{A}$. On the image classification task, we use $\mathcal{A}$ to explain 15 models and get 15 attribution results respectively. Then we aggregate the 15 attribution results through normalization and average and obtain the pseudo ground truth. After that, we measure the similarity score between the pseudo ground truth and each of the 15 attributions, as the PScore for the evaluation result of $\mathcal{A}$ on each model. In this way, the faithfulness evaluation of $\mathcal{A}$ using the PScore metric is done.

(2) This metric requires some assumptions and preconditions, that

- The models in $\mathcal{M}$ should be well-trained, otherwise both the predictions and attributions can be random and bad. According to our experiments, ImageNet-pretrained models released in public are safe to use for image classification tasks.
- The number of models should be large. The original paper [28] suggests using 15 models, while we use 9 models for image tasks and 6 models for text tasks for reducing the computation complexity.

**Synthetic Ground Truth**. Synthetic datasets with sophisticated designs similar to "data poisoning" or "adversarial attacks" can provide synthetic ground truths of explanations [32]. Note that the "attacks" we intentionally apply here are noticeable and describable, simply consisting of painted patches. The idea is that the synthetic patches on the images serve as supervision signals for training models. Labels of the images with synthetic patches will be reversed. These patches constitute explanation ground truths because no other patterns can lead to correct predictions by design. Therefore, models trained on such datasets must attribute their predictions to the synthetic ground truths. To explain a well-trained model, effective feature attribution methods should produce explanations that closely match the synthetic ground truths. We propose using Average Precision (AP) and Area Under the Receiver Operating Characteristic Curve (AUC-ROC) to quantify the matching score, referred to as SynScore. Formally, SynScore can be defined as follows:

$$\text{SynScore}_{metric}(\boldsymbol{x}) = metric(Syn(\boldsymbol{x}), \mathcal{A}(\boldsymbol{x}, \boldsymbol{f})) , \tag{6}$$

where $metric$ can be AP or AUC-ROC and $Syn(\boldsymbol{x})$ is the synthetic ground truth of explanation for the sample $\boldsymbol{x}$.

These three categories of metrics are commonly used in recent work, but each has its own limitations. We do not expect that any single category accurately measures the faithfulness of feature attribution methods. Evaluating them together through cross-comparative analysis can help to understand their strengths and weaknesses. This could provide useful insights until more principled metrics emerge.

## 2.4 Benchmark Pipeline and Modular Implementations

We currently provide evaluations on two data modalities, i.e., images and texts. For images, we choose a small amount (5000) of images from ImageNet as the benchmark dataset, multiple pre-trained image classification networks as the benchmark models and multiple feature attribution methods as the benchmark explainers. Each explainer will produce an explanation given a model for each image in the dataset, and each explanation will be given a score by applying the evaluation metrics. Finally, the faithfulness of the explainer will be quantified by the average of scores across all images in the dataset. For texts it is very similar except that the base dataset is MovieReview and the base models are for the task of sentiment analysis. The pipeline is illustrated in Figure 1.

The benchmark pipeline is implemented following InterpretDL [29] in a modular-designed style and is publicly available [2]. This means that from deep models, feature attribution methods, to evaluation metrics, their implementations are independent modules. A code sample to obtain the explanation and the evaluation results is shown in Listing 1. Therefore, new methods and metrics can also be easily added and compatible across deep learning library frameworks [3]. For example, one can use Pytorch to obtain the explanations, e.g., Captum [26], and use our benchmark to do the evaluations. To show the benchmark utility, we have provided a user-case scenario of using a HuggingFace model [4] for feature attributions and faithfulness evaluations on the X-ray pneumonia classification task. See the source code in the supplementary material for details.

## 3 Experiments and Observations

Following the benchmark pipeline as described in the previous section, we have conducted evaluation experiments. In this section, we present the experimental results, where we address several interesting research questions and introduce observations that could guide future work.

To acquaint readers with our experimental results, we first present an illustrative example. We compare this example with constant and random baselines to confirm the effectiveness of all the feature attribution methods. The results show the scores obtained by applying all possible feature attributions and measuring them using all metrics on ResNet-50. Table 1 presents the results, where the constant and random baselines perform significantly worse than the other three attribution methods (i.e., GradCAM, IG and SG). Moreover, no single attribution method achieves the best performance across all faithfulness metrics, highlighting the need for our benchmark. For rigorous analysis, the statistics in the following subsections exclude the constant and random baselines.

---

[2]`https://github.com/PaddlePaddle/InterpretDL`

[3]See a simple demo using InterpretDL with Pytorch models: `https://colab.research.google.com/drive/1ZgI1ctCc2ryPkObdPgkEwQCJ1tHZCq14`

[4]`https://huggingface.co/nickmuchi/vit-base-xray-pneumonia`

```
1  import interpretdl as it
2
3  # Load a pretrained model from PaddlePaddle model zoo.
4  from paddle.vision.models import resnet50
5  model = resnet50(pretrained=True)
6
7  # Available feature attribution methods include but are not limited to
         SG, IG, LIME, BT, GA and etc. 'interpret' is the universal api.
8  algo = it.SmoothGradInterpreter(model, device="gpu:0")
9  expl_result = algo.interpret("test.jpg")
10
11 # Available faithfulness evaluation metrics include but are not
        limited to MoRF, ABPC, INFD and etc. 'evaluate' is the universal
        api. Note that some evaluators do not require the model.
12 evaluator = it.Infidelity(model)
13 eval_result = evaluator.evaluate("test.jpg", expl_result)
```

Listing 1: Codes for computing the SmoothGrad explanation and the INFD metric score, given the model of a pretrained ResNet-50 and a testing image.

Table 1: Evaluation results on the ImageNet-pretrained ResNet-50, with constant and random baselines, GradCAM, IG and SG. "Random-16" indicates that the random values are given at patch level (16×16 pixels) while "Random" is at pixel level.

| Attribution Methods | MoRF ↑ | ABPC ↑ | PScore ↑ | INFD ↓ | SynScore ↑ |
|---|---|---|---|---|---|
| Constant | 0.000 | 0.000 | N/A | 3.015 | 0.072 |
| Random-16 | 0.596 | 0.007 | N/A | 3.039 | 0.077 |
| Random | 0.599 | 0.008 | N/A | 3.015 | 0.078 |
| GradCAM | 0.628 | **0.424** | **0.835** | 2.496 | **1.000** |
| IG | **0.709** | 0.377 | 0.812 | 2.373 | 0.999 |
| SG | 0.701 | 0.369 | 0.820 | **2.323** | 0.998 |

Evaluating the faithfulness of attribution algorithms presents several challenges from three perspectives: metric-wise, attribution algorithm-wise, and classification model-wise. All of these perspectives can lead to variance in faithfulness benchmarking. We will analyze faithfulness performance from these three perspectives in the following subsections.

### 3.1 Whether There are Two Metrics that are Correlated?

Selecting appropriate metrics is critical but also challenging given the multitude of options. We introduced three families of metrics to evaluate the faithfulness of explainability methods. An intriguing research question is whether any two metrics are correlated. The short answer is yes. To investigate the inter-correlations between metrics, whether they are correlated or orthogonal, we compute the Pearson's correlation coefficient between metrics considering all possible pairs of models and explanation techniques. We can see from Figure 2 that first, the most positive correlation coefficient is located in the pair between ABPC and PScore (0.58, p-value $2.4e^{-4}$); Second, the most negative among all pairs is between INFD and PScore (-0.59, p-value $1.9e^{-4}$); Third, there is a near zero correlation for the MoRF-PScore, MoRF-INFD and PScore-SynScore pairs, indicating that the pairs are not correlated at all.

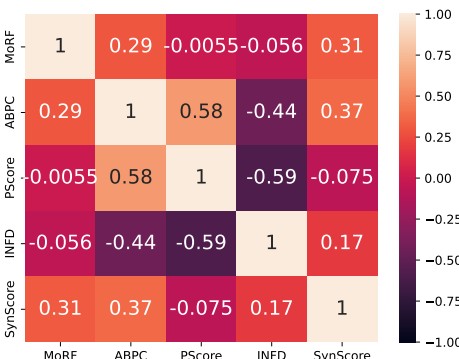

Figure 2: Correlation between metrics.

As noticed, ABPC, INFD, and PScore are potential alternatives to one another, each requiring a good amount of computation from different perspectives. Besides one trial of explanation algorithm, ABPC needs tens of forward passes of the original model. INFD requires generating random masks, which

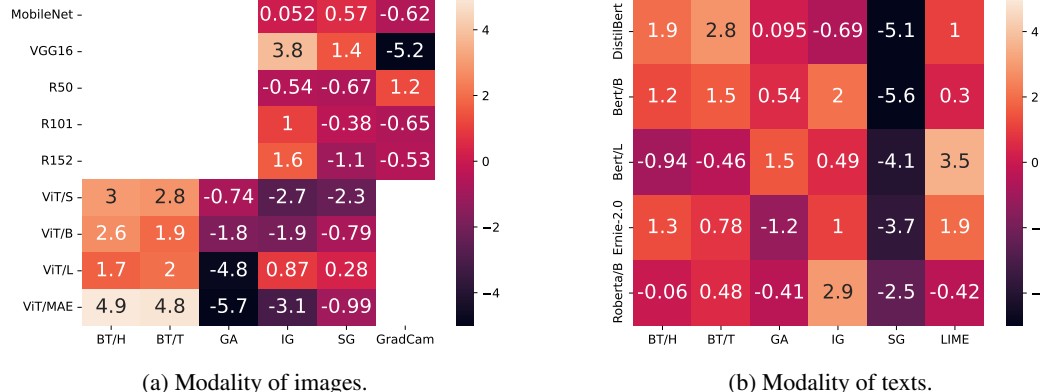

|           | BT/H  | BT/T  | GA    | IG    | SG    | GradCam |
|-----------|-------|-------|-------|-------|-------|---------|
| MobileNet |       |       |       | 0.052 | 0.57  | -0.62   |
| VGG16     |       |       |       | 3.8   | 1.4   | -5.2    |
| R50       |       |       |       | -0.54 | -0.67 | 1.2     |
| R101      |       |       |       | 1     | -0.38 | -0.65   |
| R152      |       |       |       | 1.6   | -1.1  | -0.53   |
| ViT/S     | 3     | 2.8   | -0.74 | -2.7  | -2.3  |         |
| ViT/B     | 2.6   | 1.9   | -1.8  | -1.9  | -0.79 |         |
| ViT/L     | 1.7   | 2     | -4.8  | 0.87  | 0.28  |         |
| ViT/MAE   | 4.9   | 4.8   | -5.7  | -3.1  | -0.99 |         |

(a) Modality of images.

|             | BT/H  | BT/T  | GA    | IG    | SG    | LIME  |
|-------------|-------|-------|-------|-------|-------|-------|
| DistilBert  | 1.9   | 2.8   | 0.095 | -0.69 | -5.1  | 1     |
| Bert/B      | 1.2   | 1.5   | 0.54  | 2     | -5.6  | 0.3   |
| Bert/L      | -0.94 | -0.46 | 1.5   | 0.49  | -4.1  | 3.5   |
| Roberta/B   | 1.3   | 0.78  | -1.2  | 1     | -3.7  | 1.9   |
| Ernie-2.0   | -0.06 | 0.48  | -0.41 | 2.9   | -2.5  | -0.42 |

(b) Modality of texts.

Figure 3: Averaged metric scores. A higher value indicates better faithfulness. The blanks indicate that the algorithm in a vanilla style is not suitable for the model.

is sampled from a very large space, especially for images. Meanwhile, PScore necessitates trained models' availability, with each model passing the explanation algorithm once. Practitioners should choose the most appropriate method based on the availability of models and computational resources.

There are two other key observations worth discussing. First, MoRF is weakly correlated with ABPC, with a correlation coefficient of 0.29, indicating that they are not measuring exactly the same characteristic. As discussed by Samek et al. [42], MoRF focuses only on the most important features, while ABPC also considers the ranking of the least important features. Therefore, if the goal is to filter out irrelevant features, ABPC scores should be more heavily weighted. Second, we found that ABPC, PScore, and INFD are strongly correlated with each other. In contrast, MoRF and SynScore evaluate faithfulness from different perspectives than the former three metrics.

## 3.2 Which Explanation Algorithm Demonstrates the Best Faithfulness?

Another interesting question is determining which explanation algorithm is the most faithful. Depending on the models and faithfulness metrics used, the optimal algorithm may differ. However, we still can draw several useful and instructive conclusions.

To simplify the comparison of faithfulness across explanation algorithms, we aggregate the multi-dimension metrics into a single one. First, we make the assumption that the metrics in our benchmark measure the faithfulness from different aspects since no pairs are perfectly correlated. Then we can propose a single score by averaging all of the metrics while negating the scores of INFD which is the only one ranking the faithfulness in a descent order. Moreover, we do the standardization for scores of all metrics within each model to balance the contributions of metrics. Specifically, the averaged metric is defined as:

$$\text{AvgScore} = z(\text{MoRF}) + z(\text{ABPC}) + z(\text{PScore}) - z(\text{INFD}) + z(\text{SynScore}) , \qquad (7)$$

where $z(s) = (s - \bar{s})/\sigma(s)$ is the standardization within each model. Following the formulas, we compute the averaged faithfulness score for each model-algorithm pair and show the results in Figure 3. We provide the results of each metric in the supplementary materials.

We summarize the observations as follows:

- Overall, IG generally outperforms SG except for Vision Transformers. For Transformers in NLP, SG is not an optimal choice. One possible reason is that SG adds noise in the embedding layer, and the noise scale is difficult to tune. Though further investigation is needed, we believe that the theoretical guarantee (IG satisfies the Completeness axiom, i.e., the attributions add up to the difference between the output of the model at the input and a chosen baseline) may be one of the reasons to support wide applications.

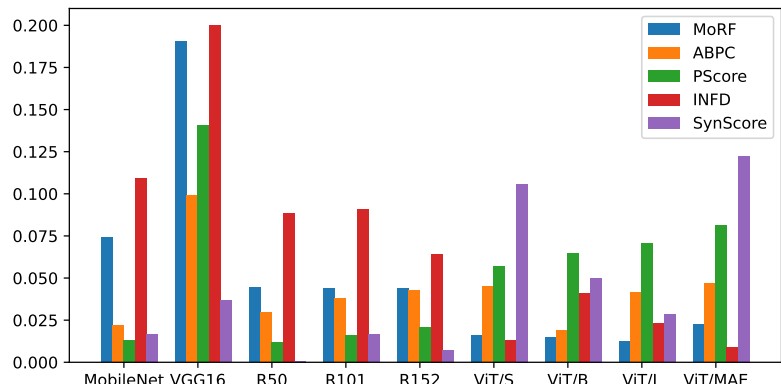

Figure 4: Sensibility measured by standard deviations across attribution methods. Note that the std for VGG16-INFD is very high (86.5) whereas we set the limit to 0.2 for better visualization.

- LIME steadily gets a high averaged score for NLP Transformers[5]. One reason is that the algorithmic computation of LIME overlaps *in some degree* with the evaluation of MoRF and ABPC. This is more obvious if we see the results of MoRF and ABPC, which are in the supplementary materials. Nevertheless, LIME is among the best algorithms measured by other metrics as well.
- For attention-based networks, including both ViTs and NLP Transformers, BT generally demonstrates higher faithfulness than others. This may stem from its accurate approximation of Transformer computations. This observation coincides with the implication of the first observation, motivating future work to design explanation algorithms through mathematical analysis of network structure, e.g., attention modules in Transformers.
- Even within the same network structure (here we have three sets of model families: ResNet-{50,101,152}, ViT-{Small,Base,Large} and Bert-{Distil,Base,Large}), no algorithm consistently achieved high faithfulness. For example, IG performed well on R101 and R152 but not R50. Although BT demonstrated the highest faithfulness in most cases, it did not do so for Bert/L. Developing a faithful algorithmic technique that works across different metrics, modalities and models remains an open challenge.

### 3.3 Which Model is the Most (In)sensitive to Explanation Algorithms?

The model's complexity and interpretability is also a key factor influencing the benchmarking performance. From the extensive evaluation results, we further investigate the sensitivity of the models to attribution methods. Given a model and a metric, we calculate the standard deviation of all metric scores across possible attribution methods. Take ResNet-50 as an example. For each evaluation metric, a standard deviation is computed among GradCAM, IG, and SG. The results are shown in Figure 4 for image classification models and Figure 5 for NLP models.

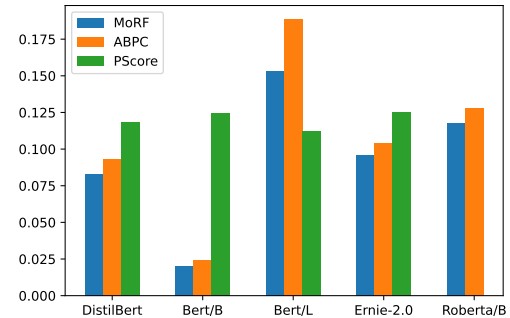

Figure 5: Sensibility measured by standard deviations across attribution methods. Since the vocabulary of Roberta/B is different from other models, thus PScore is excluded.

A model that is insensitive to attribution methods may indicate that the model can be easily explained, or all attribution methods fail to explain the model. Fortunately, the latter case does not occur in our experiments, because the attribution methods we selected for the models are relatively suitable. So, with the sensitivity, we can roughly estimate the difficulty of explaining a model. This may be helpful when designing novel network structures.

---

[5]For fair comparison, we did not involve the results of LIME in Figure 3 because the implementation of LIME involves a procedure of superpixel segmentation for reducing the computation complexity.

Although the most insensitive model is not easy to identify, the most sensitive one is VGG16, which gets the highest sensitivity in almost all metrics. For network families, we can find that ResNets get higher sensitivity than ViTs in MoRF and INFD but lower in PScore and SynScore. In either network family, the sensitivity does not vary much. As for NLP models, they have similar sensitivity as well, except Bert/B in the metrics of MoRF and ABPC. A similar observation is found in ViT/B by the ABPC metric. Existing attribution methods often use ViT/B or BERT as the primary model and achieve good evaluation results. However, some methods may work especially well for ViT/B or BERT/B but not as well for other models. Therefore, we encourage the research community to evaluate attribution methods on a variety of models with different network architectures.

## 4 Related Work

Existing work has developed benchmarks for evaluating and comparing explainability approaches. For example, Rathee et al. proposed BAGEL, a benchmark for evaluating explanation methods on graphical neural networks [39]. OpenXAI provides an open-source framework for evaluating post hoc explanation methods in tabular data [3]. Similarly, other benchmarks focus on NLP models, such as [14, 52]. The XAI-Bench library benchmarks feature attribution methods on synthetic datasets [33]. Chou et al. proposed a benchmark for counterfactual explanation methods on tabular data [11]. However, these benchmarks are limited to specific data modalities and explanation methods. A benchmark that considers multiple data modalities and explanation paradigms is still lacking. Our work addresses this gap by proposing a unified benchmark for explainable AI across different modalities, with the goal of facilitating holistic progress in the field of XAI.

## 5 Discussions, Limitations and Future Work

In this section, we provide discussions and limitations on the proposed benchmark $\mathcal{M}^4$. We also present our plans of future work for addressing the limitations.

In addition to including a wide range of feature attribution methods, faithfulness evaluation metrics, data modalities, and deep models, our benchmark $\mathcal{M}^4$ has two other properties. **The first one is efficiency and facility.** The benchmark $\mathcal{M}^4$ utilizes subsets of public datasets and evaluates the same datasets for each modality, i.e. ImageNet and MovieReview. Moreover, the models used during evaluation consist of publicly available pre-trained model weights, avoiding training new models from scratch, except for fine-tuning required in sentiment analysis. **The second one is objectiveness.** The $\mathcal{M}^4$ pipeline is objective and performed by evaluation algorithms because the faithfulness evaluation depends only on attribution methods and deep models.

We would also like to distinguish between faithfulness and interpretability. Interpretability refers to the alignment between the explanations of a model and human understanding [25]. Faithfulness is a prerequisite for interpretability and refers to how well an explanation reflects the model's functioning. This paper focuses on evaluating the faithfulness, specifically of feature attribution methods, across metrics, models, and modalities. We do not focus much on the interpretability of deep models in this work but our benchmark can be easily extended to its evaluations, e.g., with the help of ground-truth labels of image segmentation [19] and language reasoning [14].

We present the limitations and plans in the future work.

**(1)** The evaluation metrics in XAI contain several others beyond faithfulness, e.g., interpretability, sparsity, stability *etc*, while the current version of $\mathcal{M}^4$ only focuses on the faithfulness. For the comprehensive applicability of the XAI benchmark, we will progressively integrate other evaluation metrics for feature attributions. Some are easy to be plugged in. For example, the sparsity can be directly computed via the entropy of normalized attributions, but for the reason that the sparsity is not one of the faithfulness metrics, it is thus not reported in the current version of $\mathcal{M}^4$ benchmark. Another reason that we do not involve other aspects in the benchmark and focus on the faithfulness evaluations of feature attributions, is that we believe that based on faithful explanation results, we can more easily and accurately analyze other aspects of XAI. In the future, various metrics will be included in $\mathcal{M}^4$, contributing $\mathcal{M}^4$ for more comprehensive applicability in XAI.

**(2)** Our benchmark $\mathcal{M}^4$ did not include many good attribution methods, such as Shapley values based methods [35, 10], CAM variants [51, 36, 23], LIME variants [58, 30], LRPs [5, 49] attention-

based [1] and many others. However, explanations are of great variety. Although many advanced explanations beyond feature attributions have been proposed to facilitate deeper understanding of deep neural networks, their faithfulness is difficult to evaluate *a posteriori* and would be evaluated *ad hoc*. As an initial stride toward establishing a benchmark for evaluating XAI methods, concentrating on feature attribution methods would be an attainable endeavor, albeit one that still presents its own set of challenges.

**(3)** Language models and their explanations are only evaluated by the task of sentiment analysis. Though we are interested in the explanation faithfulness instead of language models' capacities and the sentiment analysis is one of the accessible tasks for faithfulness evaluations, it would be comprehensive to evaluate on other NLP tasks, e.g., those from the GLUE benchmark [50, 14, 57].

**(4)** Our benchmark $\mathcal{M}^4$ contains currently the image and text modalities. One of the future directions is to enhance the benchmark by integrating more data modalities, such as graphs, audio clips, tabular data, and multi-modality, whereas several pioneering studies [6, 9] have explored multi-modal explanations using some of the evaluation metrics in our benchmark pipeline.

**(5)** Social impacts and ethics. Our framework can assess bias and fairness issues of DNNs in high-stake applications involving sensitive attributes like gender, race, and age. This is a challenging topic and would be investigated in our future research.

## 6 Conclusions

Although existing benchmarks have advanced XAI in specific domains, a universal benchmark is lacking to compare explanation methods between models and modalities. Our work aims to address this gap by evaluating feature attribution methods on computer vision and NLP tasks using a variety of metrics. In our benchmark, we evaluated nine models using six of the most common explanation methods (LIME, SG, IG, GradCAM, GA and BT) on two modalities (image and texts) based on five evaluation metrics. We gain several observations that can inform the future design and applications of the XAI method. For future work, we plan to expand the benchmark to other modalities, including but not limited to graphs and audio. We also plan to incorporate additional evaluation perspectives, such as interpretability, stability, sparsity, *etc*.

## Acknowledgments

Xuhong Li and Haoyi Xiong were supported in part by the National Key R&D Program of China under the gradnt No. 2021ZD0110303.

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
