# OpenReview forum: "$\mathcal{M}^4$: A Unified XAI Benchmark for Faithfulness Evaluation of Feature Attribution Methods across Metrics, Modalities and Models"
_NeurIPS.cc/2023/Track/Datasets_and_Benchmarks — NeurIPS 2023 Datasets and Benchmarks Poster_

### Official Review · Reviewer_65De · 2023-07-18
**Neat combination of AI/XAI tools but how much time does the framework save for a new user?**

**Rating:** 4
**Confidence:** 4

**Strengths:**

+ allows easy comparisons of different modes, metrics, and methods
+ uses state-of-the-art models for images and text
+ few benchmarking approaches have text compared with images
+ interesting discussion on different types of metrics

**Additional Feedback:**

The paper makes a contribution to the field by providing an easy to use framework to evaluate modalities, metrics, models, and methods. The paper would have been more valuable if some of the different components were not commonly available in standardized libraries already. Most of the methods are already implemented in Captum and it is straightforward to switch to some newer pre-trained models. It is therefore unclear to a reviewer how much time this framework saves compared with performing a similar evaluation using Captum + some coding by one self. If the savings are not substantial it may be better to just use the well established Captum framework.

**Clarity:**

The paper is well written and easy to understand. Figures have been used appropriately to illustrate the main steps of the M4 framework.

**Correctness:**

The claims made in the submission appear to be correct. The dataset in the submission is collected in and sound manner and the methods and evaluation are appropriate and available online.

**Documentation:**

The submission appears to be well documented. However, the submission states that the data is proprietary and no license is listed on the submission checklist.

**Ethics:**

No.

**Limitations:**

The authors have adequately discussed some limitations of their work and outlined future directions for improvement.

**Opportunities For Improvement:**

- The explanation techniques are outdated (How about Guided-IG, AGI,  Guided Grad Cam). The value add would have been higher if some newer techniques that are not easily available in Captum were integrated.
- Within XAI for images, the most popular metrics include insertion/deletion techniques, which are not mentioned in the paper.
- Most of the models, methods, and modalities are available off-the shelf and can be easily combined by a user. While the framework makes it easier to perform evaluations, the effort needed by a user to replicate the work using Captum for image classification would be limited.
- It is unclear what type of users that would broadly adopt this framework

**Relation To Prior Work:**

Yes, the authors have contrasted their work with previous XAI frameworks such as XAI-Bench, BAGEL, and OpenAXI.

**Summary And Contributions:**

The M4 framework allows users to evaluate the faithfulness of explanations with respect to different modalities (text + images), models (ResNets + Transformers + MobileNets), methods (gradients, IG, SG), and metrics (no ground truth, pseudo ground troth, synthetic). The framework reports all the metrics for the selected modality, models, and methods. The paper used ImageNet and Movie review for the datasets. A broad variety of state of the art models are used for both the image classification and the sentiment analysis (ViT versions and different versions of BERT). In terms of explanations, standard methods such as IG,  SG, and GradCam are used. The evaluation metrics are divided into no ground truth, pseudo ground truth, and synthetic ground truth. The no ground truth metrics are based on different perturbation schemes. The paper is concluded with experiments and observations of the performance of different explanations (methods) for different experimental setups.

---

> ### Author Response · Authors · 2023-08-21
> **Response for Reviewer 65De**
>
> We appreciate your time and effort in reviewing our paper. Our responses are as follows.
>
> ## Cutting-Edge Explanation Algorithms
>
> Maybe we can start by recalling the choice of feature attribution methods in $\mathcal{M}^4$, following two considerations:
> The first one is the chronology.
> We have considered the classic ones, including LIME, IG, SG and GradCAM, which are proposed before 2018; and the SOTA ones, i.e., GA and BT, which are proposed relatively new (in 2021 and 2022 respectively).
> The second one is the range.
> We have considered various algorithms, including model-agnostic explanations (LIME), gradient-based (IG and SG) and model-specific (GradCAM for CNNs, GA and BT for Transformers).
> In this way, we demonstrate that our benchmark can be applied to every category of feature attribution algorithms.
>
> Algorithms, such as Guided-IG, AGI, Guided GradCAM and LRPs, are the same category as gradient-based and model-specific ones.
> We will try our best to add some of them in the future but enumerating all advanced algorithms would be computationally costly.
>
> ## Added Value Considering the Existence of Captum
>
> Without a doubt, Captum is a good library for XAI.
> However,
>
> - (1) As far as we know, Captum does not provide faithfulness evaluation metrics besides infidelity and sensitivity, and it does not provide benchmark baselines.
>
> - (2) (still AFAWK) new algorithms such as Transformers-specific attribution methods are not implemented by Captum yet.
>
> - (3) InterpretDL is compatible to Pytorch models now. See [this Pytorch demo with InterpretDL](https://colab.research.google.com/drive/1ZgI1ctCc2ryPk0bdPgkEwQCJ1tHZCq14?usp=sharin) as an example. We have also added a case study example of explaining [a HuggingFace model](https://huggingface.co/nickmuchi/vit-base-xray-pneumonia) for X-ray pneumonia classification, including the evaluations, in the updated source code.
>
> - (4) Moreover, the proposed benchmark and InterpretDL both follow the modular implementations. This means that the deep model, the explanation and the evaluation are separated. One can use Captum to obtain the explanations, but use our benchmark to do the evaluations. Note that the implementations for evaluations only require changes in the model's forward function and the input process. See the HuggingFace example for details. In such a way, we can at maximum reduce the migration cost.
>
> Hope this clarification can help the reviewer to dispel the misgivings.
>
> ## About the Insertion & Deletion
>
> We acknowledge that we did not discuss the Insertion&Deletion evaluation algorithm, which we should have and will be added.
> The Insertion&Deletion is analogous to LeRF and MoRF and they are based on the same underlying theory, i.e., important input features will noticeably degrade the predicted probability of a model.
> Their difference only resides on a negative sign (if we keep using for both $f(x^{(k)})$ as the perturbed input whose top-$k$ features are masked), i.e.,
>
> $\text{MoRF} (x) = \frac{1}{L+1} \sum^{L}_{k=0} (f(x^{(0)}) - f(x^{(k)}))$,
>
> and
>
> $\text{Del} (x) = \frac{1}{L+1} \sum^{L}_{k=0} f(x^{(k)}) = f(x^{(0)}) - \text{MoRF}(x) $.
>
> The same applies to the comparison between Insertion and LeRF.
> Note that we only reported the results on MoRF and ABPC (MoRF - LeRF), instead of MoRF and LeRF, following the original paper.

---

> > ### Comment · Reviewer_65De · 2023-08-29
> >
> > I would like to thank the authors for their response to my review. I have read the paper again, the other reviewers comments, and the authors responses. Unfortunately, I am still concerned about the overall usefulness of this framework to the research community.
> >
> > The main innovations within the framework:
> > - Transformers: The easy use of transformer models and attribution metrics developed for transformer models.
> > - Synthetic grounds truths: Using adversarial attacks to generate a ground truth is quite clever.
> >
> > Suggestions for improvement
> > - Easy to replicate: The datasets and most of the models are publicly available and easy to setup and use. Many of the attribution methods could be imported from Captum. The metrics are easy to implement and imported from previous work (or with minor modifications).
> > - Multi-modal: The framework provide explanations for both text and images. With the rise of multi-modal data and learning using CLIP, BLIP, etc, the authors could have considered including cross-modal explanations. For example, text explaining images and image explaining text. This would have unified the framework and improved the perceived value to this reviewer.
> > - The pseudo ground truth: The pseudo ground truth explanations idea is not very sound. If we are attempting to develop a "new and better" attribution algorithm, the pseudo ground truth will encourage the new algorithm to make the same mistakes as the previous "old and bad" algorithms.

---

> > > ### Author Response · Authors · 2023-08-30
> > > **Reply to the concerns and dispel the misunderstandings**
> > >
> > > Thanks for your time and feedback.
> > > We are glad that you acknowledged two main innovations of our proposed framework.
> > > Here, we would like to reply to the concerns and dispel the misunderstandings.
> > >
> > > ## Being Easy to Replicate and Usefulness of the Framework
> > >
> > > We would like to emphasize that the benchmark, including the attribution algorithm and the evaluation method, is designed by the modular implementation.
> > > This means that the core algorithm is separated from and independent of the deep learning model for both the attribution and evaluation methods.
> > >
> > > - **Models**. Every type of deep learning models (PyTorch, Tensorflow, PaddlePaddle, HuggingFace etc.) can be easily integrated into our benchmark pipeline with minor modifications (formatting the inputs that's all). Captum only supports the models of the original PyTorch. Nowadays more and more practitioners use HuggingFace models. It requires moderate and maybe large modifications for Captum to support HuggingFace models. However, our framework only needs minor modifications thanks to the modular implementation. See `benchmark-it-source/hf_support` and `benchmark-it-source/medical_image_example.ipynb` in the source code for example, where, in fact, a few lines of codes are modified. We believe this can benefit the community more than previous libraries, which strongly couple with one single framework.
> > >
> > > - **Algorithms**. The modular implementation moreover enables the ease to add new algorithms and new evaluation metrics, because one can focus on the core algorithm, instead of the coupling between the model and the attribution/evaluation algorithm. Moreover, most classic and commonly-used attribution algorithms and evaluation metrics that can be imported from Captum and other libraries have already been implemented by InterpretDL, as well as those Transformer-based ones, which are not supported by Captum yet.
> > >
> > > In summary, we believe that our framework is already being the form that is easy to replicate and useful to the community.
> > >
> > > ## Supports for Multi-modal Attributions
> > >
> > > Thanks for this constructive suggestion.
> > > Two Transformer-based attribution algorithms GA and BT, and some variants of classic ones such as GradCAM, support CLIP and BLIP.
> > > Part of evaluations have been performed by these two works, and this can be easily performed by our benchmark pipeline.
> > > However, for the time limit, we can not promise to finish these experiments within the rebuttal deadline, while we will add them on the camera-ready version.
> > >
> > > > GA: Hila Chefer, Shir Gur, and Lior Wolf. Generic attention-model explainability for interpreting bi-modal and encoder-decoder transformers. In Proceedings of the IEEE/CVF International Conference on Computer Vision, 2021
> > > > BT: Jiamin Chen, Xuhong Li, Lei Yu, Dejing Dou, and Haoyi Xiong. Beyond intuition: Rethinking token attributions inside transformers. Transactions on Machine Learning Research, 2022.
> > >
> > >
> > >
> > > ## The Pseudo Ground Truth Metric
> > >
> > > First, we would like to clarify that the pseudo ground truth is based on an ensemble of models for one attribution algorithm, instead of an ensemble of attributions for one model.
> > >
> > > For example, we would like to evaluate the faithfulness of **a new attribution algorithm A**.
> > > On the image classification task, we use A to explain 15 models and get 15 attribution results respectively.
> > > Then we aggregate the 15 attribution results through normalization and average and obtain the pseudo ground truth.
> > > After that, we measure the similarity score between the pseudo ground truth and each of the 15 attributions, as the PScore for the evaluation result of A on each model.
> > > In this way, the faithfulness evaluation of A using the PScore metric is done.
> > > Hope this clarification is helpful and we will add this explanation to the camera-ready version.
> > >
> > > However, this metric indeed requires some assumptions and preconditions, that
> > > - (1) The models should be well-trained, otherwise both the predictions and attributions can be random and bad. The similar situation you mentioned would happen.
> > > - (2) The number of models should be large. The original paper suggests using 15 models, while we use 9 models for image tasks and 6 models for text tasks for reducing the computation complexity.
> > >
> > > Thanks for bringing this issue to us.
> > > We will add detailed explanations and discussions on using this metric.

---

### Official Review · Reviewer_inEz · 2023-07-21
**A Unified XAI Benchmark for Faithfulness Evaluation of Feature Attribution Methods across Metrics, Modalities and Models**

**Rating:** 7
**Confidence:** 4

**Strengths:**

1. **Significance of Contribution:** The submission addresses a crucial challenge in the field of Explainable Artificial Intelligence (XAI), which is the evaluation of faithfulness in post-hoc feature attributions from deep learning models. The problem is significant, given the increasing application of deep learning models in high-stake fields like healthcare and criminal justice, where explainability is essential.

2. **Relevance to the Broader Research Community:** The proposed $M^4$ benchmark has wide applicability across various data types (image and text) and network architectures. It can be a valuable resource for researchers and practitioners in the AI community, particularly for those interested in XAI.

3. **Quality of Research:** The taxonomy proposed in the submission, which categorizes XAI evaluation metrics based on their ground truth requirements, shows an innovative and thoughtful approach to the problem. It also introduces a systematic method to understand and compare different evaluation metrics. The authors conduct extensive experiments to test and validate their framework, which contributes to the robustness and validity of the research.

4. **Ethical and Social Implications:** This research has important ethical implications as it helps in increasing the transparency of deep learning models. By evaluating and improving the faithfulness of explanations provided by XAI techniques, the proposed benchmark could potentially make AI more understandable and trustworthy, particularly in high-stake applications. It could also mitigate the risks associated with unfaithful explanations, such as deceptive insights, which could lead to incorrect decisions and actions.

5. **Open Accessibility:** The authors have made their work available to the public, contributing to the ethos of open research and potentially promoting further innovation in the field. This allows the broader community to review, reproduce, and extend their work, enhancing the reliability and reach of their contributions.

**Additional Feedback:**

1. Benchmark scope: The authors have made an ambitious effort to create a benchmark covering multiple data modalities and explanation paradigms. However, the $M^4$ benchmark does not currently cover all notable explanation methods. It might be beneficial to provide a more detailed explanation about the selection criteria for the inclusion of certain methods over others. Additionally, future extensions to the benchmark should aim to include other prevalent attribution methods like CAM variants, LIME variants, LRPs, and attention-based methods. The reasons for their current omission and a roadmap for their future inclusion would be useful.

2. Benchmark utility: The $M^4$ benchmark aims to be useful for practitioners and researchers alike. However, more explicit guidance or examples on how to use the benchmark for model development, selection, or auditing tasks would increase its utility for potential users. Perhaps a use-case scenario or a real-world application case study might help to demonstrate the practical implications of the benchmark.

3. Experimental setup: Detailed information about the specific setup used in the evaluation process would aid in replicability. For instance, how were the hyperparameters for each method chosen? How was fine-tuning performed for sentiment analysis?

4. Interpretability vs. Faithfulness: The authors point out that their work focuses on evaluating faithfulness rather than interpretability of models, though they acknowledge the two are closely linked. Given the importance of interpretability in practical applications of XAI, it might be helpful to discuss how the benchmark could be adapted or extended to better assess interpretability in the future.

5. Complexity and accessibility: One potential drawback of a comprehensive benchmark like $M^4$ could be its complexity. For users who are not experts in XAI, this benchmark could be daunting. Providing a user-friendly interface or clear tutorials on how to use the benchmark might be beneficial.

6. Implications of findings: The authors have presented interesting findings about the performance of different methods across models. It would be helpful to include more discussion on the implications of these findings for the field, and how they might guide the development of future XAI methods. For example, what are the implications of IG generally outperforming SG except for Vision Transformers, or BT demonstrating higher faithfulness for attention-based networks?

Overall, the paper presents a valuable contribution to the XAI field with the $M^4$ benchmark. However, expanding upon these points can further improve the clarity and usability of the work.

**Clarity:**

Yes, the paper appears to be well-written overall. The authors have clearly explained the purpose and significance of their research, the methods used, and their findings. They have also thoughtfully discussed the implications, strengths, and limitations of their work.

Key concepts are clearly defined, such as the distinction between faithfulness and interpretability. The arguments are logically structured, and the flow of ideas is coherent, moving from the context and background of the study, to the research problem, methodology, results, and implications.

The authors have effectively used diagrams and formulas to support and clarify their explanations, such as the illustration of their experimental results and the formula for computing the averaged faithfulness score. They have also provided a clear and succinct summary of their contributions.

The language is formal and appropriate for a scientific paper, and the technical terms and acronyms are adequately explained, which makes the paper accessible to readers with different levels of familiarity with the topic.

**Correctness:**

It appears that the authors have made valid and sound claims about their work. They have carefully designed experiments to compare and evaluate the performance of various feature attribution methods across different models using their $M^4$ benchmark.

The evaluation method applied by the authors seems appropriate and has been carried out correctly. They use a range of different metrics and have developed an averaged score to assess faithfulness across multiple dimensions. Their approach allows them to evaluate the effectiveness of different explanation algorithms in a nuanced manner, taking into account the different aspects of faithfulness. The authors also examine correlations between different metrics to ensure a comprehensive evaluation.

The authors further recognize that the optimal explanation algorithm can vary depending on the models and faithfulness metrics used, which shows a sound understanding of the complexity of the problem they are addressing. They make an effort to simplify the comparison across explanation algorithms by assuming that the metrics in their benchmark measure faithfulness from different aspects since no pairs are perfectly correlated.

The experimental design also includes comparison with constant and random baselines to validate the effectiveness of the feature attribution methods under consideration. The authors' analysis appropriately distinguishes between different perspectives including metric-wise, attribution algorithm-wise, and classification model-wise perspectives. A small suggestion for Table 1: Evaluation results on the ImageNet-pretrained ResNet-50 is to consider running multiple random seeds and report the average results.


**Documentation:**

Yes, there is sufficient detail provided to support reproducibility for the benchmark proposed by the authors. They have detailed the approach they have taken to develop the M4 benchmark, including the categorization of faithfulness metrics, the methods of evaluation across different modalities (images and texts), and the use of various deep learning models.

However, the authors do not provide explicit details about a hosting, licensing, and maintenance plan for the M4 benchmark. It is essential for these aspects to be clearly communicated to ensure long-term accessibility and utility of the benchmark.

Furthermore, while the authors mention using subsets of public datasets like ImageNet and MovieReview, there's no detailed discussion around the ethical and responsible use of these datasets. It's important to consider such factors, especially when the datasets are being used for high-stakes applications like healthcare, criminal justice, and law, which are contexts mentioned in the paper.

For further improvement, it would be beneficial if the authors could provide a URL for direct access to the benchmark, elaborate on their plans for maintaining and updating the benchmark, and address any potential ethical considerations related to the use of their benchmark and the datasets used.

**Ethics:**

Based on the provided information, there doesn't appear to be any immediate ethical concerns with this submission. The work focuses on the development of a benchmark, $M^4$, for evaluating explainable AI (XAI) methods and doesn't involve new data collection, thus direct privacy or consent issues are unlikely to arise.

However, a few points that the authors might need to consider are:

1. Dataset origin and use: The authors mention the use of subsets of public datasets like ImageNet and MovieReview. If these datasets contain personal or sensitive information, the authors should ensure that they're complying with all relevant data protection and privacy laws. Moreover, it would be helpful if the authors discuss any ethical implications associated with the use of these datasets.

2. Responsible use of the benchmark: The authors have designed the M4 benchmark to evaluate the faithfulness of XAI methods, particularly for high-stakes applications like healthcare, criminal justice, and law. They should consider providing guidelines for the responsible use of their benchmark to prevent misuse or misinterpretation of results that could potentially have adverse societal impacts.

3. Inclusion of diverse data types and models: The M4 benchmark proposes to cover diverse data types and models. In doing so, it is essential to ensure that the models and data do not exhibit or perpetuate bias, as biased AI systems can lead to unfair or discriminatory outcomes.

In summary, while there aren't any explicit ethical concerns raised by the submission, the authors should ensure they've considered these potential issues and have taken the appropriate steps to address them.

**Limitations:**

The authors have shown a commendable level of self-awareness in discussing the limitations of their work. They have identified three main limitations:

1. The current version of the $M^4$ benchmark focuses solely on the evaluation of faithfulness, although other evaluation metrics in XAI such as interpretability, sparsity, stability, etc., exist. This focus could limit the benchmark's comprehensive applicability in the field of XAI.

2. The benchmark did not include many other well-established attribution methods, such as CAM variants, LIME variants, LRPs, attention-based methods, etc. This limits the range of techniques that can be tested and compared using this benchmark.

3. Language models and their explanations are only evaluated through sentiment analysis, limiting the breadth of NLP tasks covered. Other NLP tasks from the GLUE benchmark, for instance, were not considered in the evaluations.

While the authors have highlighted these limitations, they have not explicitly discussed the potential negative societal impact of their work. For improvement, they could consider discussing the implications of using an incomplete set of evaluation metrics or attribution methods in their benchmark. They could also delve deeper into the potential repercussions of only using sentiment analysis for evaluating language models, such as potentially biased or ungeneralizable results.

Further, the authors could suggest potential mitigation strategies for these limitations. For instance, they could propose plans to extend the benchmark to include more attribution methods or a broader range of NLP tasks. They could also discuss how other evaluation metrics could be integrated into the $M^4$ benchmark, thereby ensuring a more comprehensive evaluation of XAI techniques.

Lastly, given the focus on the explainability of AI systems, it would also be beneficial for the authors to engage with the ethical dimensions of their work, such as how their research might contribute to the transparency and fairness of AI systems and their societal implications. They could discuss how their benchmark could be used to ensure the ethical use of AI systems and help avoid potential misuse or harms.

**Opportunities For Improvement:**

1. **Significance of Contribution:** While the contribution is significant in providing a benchmark and taxonomy for XAI evaluations, it appears to focus mostly on post-hoc feature attributions. There are other important aspects of XAI, like model transparency or interpretability of the training process, which aren't addressed in this submission. The benchmark's utility might be limited for researchers working with these other areas of XAI.

2. **Relevance to the Broader Research Community:** While the $M^4$ benchmark covers multiple data types and network architectures, it may not be comprehensive enough to cater to the diverse needs of the broad AI research community. The utility of the benchmark could be limited for emerging or niche models, data types, or domains not covered in the current iteration.

3. **Quality of Research:** While the authors have proposed a taxonomy and conducted extensive experiments, the effectiveness of their approach might still be contingent on certain assumptions. For example, the categories of metrics are based on the availability of ground truth, but the nature of ground truth can be contentious and vary significantly across different tasks and domains. Also, the benchmark may be influenced by the quality and representativeness of the datasets used.

4. **Ethical and Social Implications:** While the $M^4$ benchmark can help in enhancing transparency of AI systems, it may not fully address other ethical concerns associated with AI, such as bias, fairness, or misuse of AI systems. The submission does not provide any explicit discussion or guidelines on these issues.

5. **Open Accessibility:** Although the $M^4$ benchmark and related materials are publicly available, their usability might be limited if they lack sufficient documentation or are complex to use. If the community cannot easily understand or adopt the benchmark, its impact could be diminished.


**Relation To Prior Work:**

Yes, the authors have clearly discussed how their work differs from previous contributions. They have provided a comprehensive overview of existing benchmarks for evaluating and comparing explainability approaches, and they have identified the limitations in these existing works.

Specifically, they noted that existing benchmarks, such as BAGEL, OpenXAI, XAI-Bench, and others, are restricted to specific data modalities and explanation methods. These benchmarks focus either on graphical neural networks, tabular data, or NLP models, and each has its own method of explanation, whether it's post hoc explanation, feature attribution, or counterfactual explanation.

The authors underscore that their work is unique in proposing a unified benchmark for explainable AI that can handle multiple data modalities, which will facilitate a more holistic understanding and progress in the field of XAI. The difference in their approach and its potential impact are clearly highlighted.

**Summary And Contributions:**

The submission presents a novel benchmark, $M^4$, for evaluating Explainable Artificial Intelligence (XAI) techniques. Recognizing the challenges in assessing the faithfulness of explanations provided by deep learning models, particularly due to model heterogeneity and lack of ground truth, the authors developed a unified platform for evaluation across multiple data types (image and text) and different network architectures (ResNets, MobileNets, Transformers).

 $M^4$ proposes a taxonomy for XAI evaluation metrics, categorizing them into three groups based on their ground truth requirements. The authors have implemented classic and state-of-the-art feature attribution methods using InterpretDL, performing extensive experiments to compare these methods and provide comprehensive benchmark baselines. They also made several interesting observations that could inform the design of future attribution algorithms.

The benchmark’s implementation and all related materials are open-sourced and publicly available at the provided GitHub link.

Contributions:

Creation of the $M^4$ benchmark for consistent evaluation of XAI techniques across different models, data modalities, and network architectures.
Introduction of a taxonomy categorizing XAI evaluation metrics based on their ground truth requirements.
Implementation and comparison of various feature attribution methods through extensive experiments.
Provision of benchmark baselines and useful insights for developing future attribution algorithms.
Open-source availability of the benchmark for wider use and further development by the community.

---

> ### Author Response · Authors · 2023-08-21
> **Response for Reviewer inEz (Part 2/2)**
>
> ## About Additional Feedback
>
> **1. Benchmark Scope**.
> Our choice of feature attribution methods in $\mathcal{M}^4$, follows two considerations:
> The first one is the chronology.
> We have considered the classic ones, including LIME, IG, SG and GradCAM, which are proposed before 2018; and the SOTA ones, i.e., GA and BT, which are proposed relatively new (in 2021 and 2022 respectively).
> The second one is the range.
> We have considered various algorithms, including model-agnostic explanations (LIME), gradient-based (IG and SG) and model-specific (GradCAM for CNNs, GA and BT for Transformers).
> In this way, we demonstrate that our benchmark can be applied to every category of feature attribution algorithms.
>
> Algorithms, such as Guided-IG, AGI, Guided GradCAM, and LRPs, are the same category as gradient-based and model-specific ones.
> We will try our best to add some of them in the future, but enumerating all advanced algorithms would be computationally costly.
>
> **Benchmark utility**
>
> We have added a user-case scenario of using [a HuggingFace model](https://huggingface.co/nickmuchi/vit-base-xray-pneumonia) for feature attributions and faithfulness evaluations on the X-ray pneumonia classification task.
> See the updated source code for the demonstration.
> From the results, we can see that IG gets higher faithfulness over SG in such cases, indicating that the IG attributions are more trustful for explaining the model.
>
> **Experimental setup**.
> These hyperparameters for finetuning models on the downstream tasks can be found in the supplementary materials.
> Those for attributions and evaluations can be found in the source code, where we also provide the direct scripts for the reproducible purpose.
>
> **Interpretability vs. Faithfulness**.
> Other aspects such as interpretability are somehow based on the faithfulness of explanation methods.
> For example, interpretability is defined as the ability (of the model) to explain or to present in understandable terms to a human.
> However, the post-hoc feature attributions are calculated by the explanation methods.
> We believe that when based on (more) faithful explanation results, we can (more) easily and accurately analyze other aspects of XAI.
>
> **Complexity and accessibility**.
> Thanks for the advice.
> Detailed documentation and getting-started tutorials will be added to the benchmark when releasing to the public.
>
> **Implications of Findings**.
> We will add some discussion of the findings.
> For example, IG generally outperforms SG for the argument that IG satisfies the Completeness axiom (the attributions add up to the difference between the output of the model at the input and a chosen baseline) while SG does not.
> This theoretical guarantee may be beneficial for wide applications.
> Similar implication for the observation of BT demonstrating higher faithfulness for attention-based networks.
> BT is designed by mathematical approximations with limited approximation errors, while the other methods are proposed with bold assumptions.
>
> ## In Summary
>
> Thanks again for the very insightful reviews and constructive suggestions.
> To recall, all modifications will be added to the revised version of the manuscript.

---

> ### Author Response · Authors · 2023-08-21
> **Response for Reviewer inEz (Part 1/2)**
>
> We appreciate your time and effort in reviewing our paper. Our responses are as follows.
>
> ## About Other Aspects of XAI (e.g., Model Transparence, Interpretability of the Training Process)
>
> We would like to address other explanation methods beyond post-hoc feature attributions and other aspects of XAI beyond faithfulness evaluations.
> However, they both face various challenges.
> First, explanations are of great variety.
> Although many advanced explanations beyond feature attributions have been proposed to facilitate a deeper understanding of deep neural networks, their faithfulness is difficult to evaluate _a posteriori_ and would be evaluated _ad hoc_.
> As an initial step toward establishing a benchmark for evaluating XAI methods, concentrating on feature attribution methods would be an achievable endeavor, albeit one that still presents its own set of challenges.
> Second, other aspects, such as interpretability, are somehow based on the faithfulness of explanation methods.
> We believe that based on faithful explanation results, we can more easily and accurately analyze other aspects of XAI.
>
> ## About Broader Research Community
>
> The proposed benchmark and the library behind InterpretDL both follow the modular implementations.
> This means that the deep model, the explanation and the evaluation are separated.
> One can plug in any model that has _forward_ and _backward_ functions or one can design these interfaces, which is the only thing to do.
> For other data types and domains, we plan to expand the benchmark to other modalities, including but not limited to graphs and audio; while the current iteration containing texts and images covers a wide community.
>
> ## Quality of Research
>
> Thanks for the advice.
> We have added discussions on the limitation by the used dataset and task.
> However, we would like to clarify that one application of our benchmark is to help practitioners in a specific domain.
> Users can apply the pipeline of our benchmark on their own tasks and datasets correspondingly, and choose the best attribution method in the specific domain.
> In such a way, our benchmark can be applied across domains.
>
> ## About Ethical and Social Implications
>
> Thanks for your advice.
> We have added discussions on social impacts and ethics in Section 5 of the revised version.
> Our framework can assess the bias and fairness issues of DNNs in high-stake applications involving sensitive attributes such as gender, race, and age. This is a challenging topic and would be investigated in our future research.
>
> ## Open Accessibility and License
>
> The documentation, including the getting-started tutorials, will be prepared when released to the public.
> The documentation will also be further improved in the future according to the users' feedback.
>
> Regarding the license, the code follows the same license as InterpretDL, i.e., [Apache License 2.0](https://github.com/PaddlePaddle/InterpretDL/blob/master/LICENSE).
> The datasets we used in our manuscript follow their original license respectively.
> As for their own datasets, users do not necessarily follow our license, except for the one for codes.
>
>
> ## Limitations
>
> Thanks for constructive suggestions on addressing the limitations, including ethics, documentations for plans of extending the boarder of the benchmark, the uses of datasets, and other aspects of XAI.
> We have revised the limitation and future work sections following your suggestions.

---

### Official Review · Reviewer_Jbv9 · 2023-07-24
**Benchmarking feature attribution methods**

**Rating:** 6
**Confidence:** 3
**Clarity:** The paper is very well written. It is…

**Strengths:**

The paper addresses an important but challenging problem, i.e., benchmarking explainable AI methods.

The choice of tasks such as image classification and sentiment analysis is reasonable.

Categorizing metrics into no ground truth, pseudo ground truth, and synthetic ground truth is reasonable.

The experiments are extensive and the observations are useful.

**Additional Feedback:**

Is it possible to have the so-called white-box models that are more easily explained than the existing models?

**Correctness:**

The claims made in the paper seem correct. The evaluation methods and experiment design are appropriate and performed correctly.

**Documentation:**

The document is sufficient. There is sufficient detail to support reproducibility.

**Ethics:**

I do not see any ethical concerns.

**Limitations:**

The paper addresses the limitations. I do not see any potential negative societal impact.

**Opportunities For Improvement:**

The main shortcoming of this paper is novelty. The paper mainly categorizes existing metrics and evaluates existing methods and models. It may be interesting to explore some theoretical frameworks such as Shapley value etc.

**Relation To Prior Work:**

The paper discussed prior work and the differences between this paper and prior work are clearly explained.

**Summary And Contributions:**

This paper benchmarks feature attribution methods for image classification and sentiment analysis, with various metrics and models, for the purpose of evaluating methods for explainable AI. The paper categorizes commonly used metrics, evaluates diverse attribution methods, conducts extensive comparisons of baselines, and obtains useful observations.

---

> ### Author Response · Authors · 2023-08-21
> **# Response for Reviewer Jbv9**
>
> We appreciate your time and effort in reviewing our paper. Our responses are as follows.
>
> ## About the Novelty
>
> We aim to address the challenges of faithfulness evaluations of XAI algorithms.
> Most previous benchmarks focus on a single data type and model, while the feature attribution methods can effectively be applied to various data modalities and different models.
> By first categorizing the existing evaluation metrics, we can systematically analyze and benchmark the feature attribution methods.
> We also propose a unified benchmark pipeline across metrics, modalities, and models.
> Extensive experiments and benchmark baselines have been provided, where several interesting observations are made.
>
> ## Discussions on Shapley Values based Methods
>
> Thanks for the suggestion.
> We will add discussions on the Shapley values based methods.
> The Shapley values based methods have high computation complexity, compared to gradient-based and other methods in our benchmark.
> It would be time-consuming to conduct evaluations across models and modalities.
>
> ## About the White-Box Models
>
> It is not necessary to include white-box models in our benchmark.
> We provide our arguments as follows.
> Let's take the linear model as an example, which is generally agreed as a white-box model.
> Different attribution methods may provide the same accurate and faithful attribution because the linear model is easy to be explained.
> In such case, the evaluation results will be the same, and cannot indicate which explanation algorithm can provide more faithful results.

---

### Decision · Program_Chairs · 2023-09-22

**Decision:**

Accept (Poster)

**Comment:**

The paper presents a new benchmark for faithfulness evaluation of feature attribution methods. Unlike exiting benchmarks focusing on a single data type and model, the proposed benchmark makes it easier to users to systematically analyze and benchmark different feature attribution methods (including model-agnostic explanations like LIME, gradient-based  approached like IG and SG, and model-specific (approaches like GradCAM for CNNs, and GA and BT for Transformers) across metrics (categorizing them into three groups w.r.t. kind of ground truth they represent), modalities (image and text), and models (ResNets, MobileNets, Transformers). The authors also introduced a unified benchmark pipeline to facilitate that. The extensive experiments are convincing.
During the rebuttal phase, the authors addressed the main concerns of the reviewers regarding the scope, utility and novelty of the introduced benchmark.
Notably, they have added a user-case scenario of using a HuggingFace model for feature attributions and faithfulness evaluations on the X-ray pneumonia classification task.

The proposed benchmark as is can further facilitate further development and understanding of feature attribution methods. Furthermore, its modular design should help to maintain and update the benchmark as the research area of XAI continues to mature and new feature attribution methods and way to evaluate them emerge.
The benchmark’s implementation and all related materials are open-sourced and publicly available at the provided GitHub.